# Public Participation in Biodiversity Impact Assessment in the State of West Bengal, India: Present Status and Finding Ways for Improvement

Rajarshi Chakraborty [1] , Andre Lindner [2,*] and Wolfgang Wende [2,3]

1 Department of Environment, Government of West Bengal, Kolkata 7000106, India; rajarshichakra@gmail.com
2 School of Civil and Environmental Engineering, Technische Universität Dresden, 01069 Dresden, Germany; w.wende@ioer.de
3 Leibniz Institute of Ecological Urban and Regional Development, Landscape Change and Management, Weberplatz 1, 01217 Dresden, Germany
* Correspondence: andre.lindner@tu-dresden.de

**Abstract:** The present status of public participation in EIA particularly concerning biodiversity in West Bengal, India was studied. The issues raised in 50 public hearings were analyzed and chapters on biodiversity in 20 EIA reports were studied. Areas needing improvement were identified. Scientific literature was studied to gather best practices/concepts. It was observed that, despite all enabling legal provisions, public participation in EIA has not grown to its full potential. The discussion was mostly on jobs and benefits (and little on biodiversity impact). EIA reports did not provide any spatial information on biodiversity-rich/sensitive areas or impact on bio-resources that are used by people. We identified four pillars of effective public participation in EIA as: (i) institutional opportunity and conducive environment for participation; (ii) interest of local people to participate; (iii) capacity building of local people; and, (iv) support of clearance process. Specific recommendations under each are provided. A simple matrix for Biodiversity Impact Assessment and a list of components for the improvement of biodiversity, for use of local people, have been developed.

**Keywords:** environmental impact assessment (EIA); biodiversity offsets; public participation; local people

## 1. Introduction

Biodiversity apart from its intrinsic value is relevant to humans for the ecosystem services that it helps to provide. From food, healthy air, water, waste decomposition, maintenance of soil fertility, climate stability, to providing mental relaxation and pleasure, biodiversity is indispensable to mankind's survival. However, human beings as dominant and prevalent species are causing changes in the ecosystem resulting in biodiversity loss. It is estimated that, out of nine planetary boundaries, humanity has already transgressed three of which the most prominent is the rate of biodiversity loss [1]. The interdependence of biodiversity elements on each other through a complex web of interactions for their survival and for providing ecosystem services makes the issue of biodiversity loss a cause of great concern.

The adverse impact on biodiversity from developmental activities can be minimized and irreversible losses can be avoided by undertaking environmental impact assessment (EIA) and taking mitigation and offsetting measures into account [2]. Avoiding biodiversity-rich areas, minimizing impact, executing restoration activities, and implementing offset measures are possible measures of the mitigation hierarchy. Public participation forms an integral part of the EIA process. It is intended to serve multiple purposes, like providing knowledge of local environment and community for incorporation into baseline data [3],

drawing attention to the concerns of local people [4], resolution of conflicts [5], and bringing the full range of options to the government [6].

West Bengal, a state within the tropical country of India, has a wide variety of geographic features and landscapes, resulting in rich biodiversity. Like the rest of India, EIA has to be done in West Bengal as per law for a range of proposed projects requiring environmental clearance. Biodiversity is one of the aspects that have to be considered in EIA study. Thus, biodiversity impact assessment (BIA) is a subset of overall EIA. Public participation is a mandatory component of the EIA process. Although public participation has been institutionalized through legal provisions, it is judicious to analyze the implementation status of this important component of the EIA process to detect gaps and find ways of improvement.

This paper intends to study the present scope (in the form of legal provision) and utilization of public participation in BIA in West Bengal, India. From the present status of implementation, it wants to detect deficiencies and identify areas that need improvement. Utilizing scientific literature on effective public participation and BIA, the paper aims to develop recommendations for further improvement in terms of wider participation and quality.

## 2. Legal Provisions for Public Participation in EIA in India

In India, there exist two separate legal provisions under which public participation in EIA for a proposed project can take place. These are stated below.

### 2.1. Environmental Impact Assessment Notification, 2006

The projects covered by the Environmental Impact Assessment Notification, 2006 (which was issued under the Environment Protection Act, 1986) have to take prior environmental clearance. As per the schedule appended to the Notification, a section of those projects have to do EIA with 'public consultation' as an integral part. Public consultation has to be done after the submission of draft EIA by the project proponent and would constitute—(i) submission of comments in writing and (ii) public hearing. The copies of the draft EIA are circulated to different government offices, where people can access the report. Because the submission of written comments is a rarity (with no written comments being received by the State Environment Impact Assessment Authority in the last five years), public hearings presently form the exclusive component of such consultation in West Bengal. Public hearings are presided over by a senior government official, and they constitute a presentation on the proposed project by the project proponent followed by questions/comments from the public. People are informed about the hearings through an announcement in two newspapers. The project proponent has to address the issues raised in the public hearing in the final EIA report.

Apart from mitigation measures of adverse impacts, the project proponent is required to undertake works around the project area for the betterment of the environment and people as part of Corporate Environment Responsibility (CER). Suggestions made by people during the public hearing would form one of the bases for choosing activities under CER. Recent Government Notification put stress on the specific activities mentioned in a public hearing (instead of allocation of funds under CER).

Thus, in public hearings, people can not only talk about the impact issues of the proposed project, but also suggest components for CER.

### 2.2. Biological Diversity Act, 2002

As per Section 36.4.i of the Biological Diversity Act, 2002, based on the likelihood of adverse impact on biodiversity, the Central Government may arrange for public participation for impact assessment of a project. However, to date, this provision has not been utilized in West Bengal.

## 3. Study of Scientific Literature

We conducted a literature search using Google Scholar, using the keywords 'public participation in EIA' and 'biodiversity impact assessment'. Because both searches generated a large number of hits, based on the relevance of the paper's title, 100 papers were collected on each topic.

After going through the abstract, based on the criteria if the paper 'pointed out constraints of participation' or/and 'provided recommendations for improved participation', 35 papers (of which 10 were reviews and rest were case studies) on public consultation in EIA/environmental management were selected. These papers were thoroughly studied to collect good practices of public consultation.

Regarding BIA and the related topics of biodiversity conservation, biodiversity loss, restoration, and offsets, a total of 65 papers (including three guidelines for doing BIA) were selected for the detailed study after reading the abstracts.

### 3.1. Public Participation in EIA

Pertinent issues affecting public participation in EIA were searched out from studied literature for use in the preparation of recommendations. These issues were the institutionalization of participation [7], use of local dialect in hearing [8], access to information [9], manipulation of participation [10], stakeholder analysis [7], the interest of people to participate [6], the capacity of people [11], and attitude exhibited by local/state/federal governments towards citizen involvement [12].

### 3.2. BIA and Biodiversity Conservation

The literature on BIA, criteria for biological conservation, and causes of biodiversity loss were studied to collect important biodiversity attributes and drivers of change for the preparation of a matrix for impact determination. Species richness, the importance for life-history stages, like migration and breeding, the presence of species of conservation concern, the occurrence of restricted-range species [13]; resilience [14]; functional diversity and response diversity [15]; mobile link species [16]; keystone species [17]; food chain [18]; and, ecosystem services [19] were found as important concepts to be considered for biodiversity conservation and impact assessment. Factors that may affect biodiversity include human transformations of land cover and land use [20], habitat loss [21], fragmentation of habitat [22]; chemical pollution, hunting, and invasive alien species [23]; and, littered plastic waste [24]. Cultural practices of local people [25] play an important role in biodiversity conservation. Eneji et al., 2009 [12] concluded that unless rural people perceive biodiversity conservation efforts to be serving their economic and cultural interests, public participation will remain low.

Papers on restoration and offset measures provided various concepts/suggestions, which include the creation of heterogeneity in landscape [23], including the creation of wild patch [26], protection of remnant small wild patches [27,28], planting a diversity of tree species along streams and roads [29], planting of native species [30], creation of habitat banks [31], conservation of landraces and agro-biodiversity [32], control of invasive species as offset measure [33], identification of non-offsetable impacts which should be prevented [34], and inclusion of ecosystem services in offsetting [35].

## 4. Methodology

### 4.1. Study of Minutes of Public Hearings

A study of meeting minutes of all 50 public hearings conducted from 28th November 2017 to 28th August 2019 in West Bengal was done to analyze the issues that were raised in those hearings. These hearings were for 50 proposed projects consisting of 36 industrial, four mining, two gas extraction, one infrastructural, and seven storage and pipelines (of hazardous chemicals) projects. The number of people who spoke in each of those hearings was noted. Comments that were made in the hearings were counted and categorized into different topics. The purpose of the study was done to gain insight into the

present utilization of opportunities of public consultation in EIA, particularly concerning biodiversity issues.

### 4.2. Study of EIA Reports

Twenty EIA reports (for 12 industrial, two infrastructural, and six building projects), which were submitted to the State Environment Impact Assessment Authority (SEIAA), West Bengal from April 2019 to December 2019 were used as a sample for the study. The sample sets of public hearings and EIA reports were different (due to differences in time and number). The biodiversity-related chapters of EIA reports were studied to find out the contents of: (a) existing biodiversity status, (b) impact assessment, and (c) proposed mitigation measures. The objective was to determine whether there was any scope for further improvement in EIA through a greater utilization of public consultation.

### 4.3. Generation of Recommendations

An analysis of the present status of people's participation in public hearings and scientific literature review was used to develop suitable recommendations for putting scientific findings in governance practices and develop tools that are suitable for even non-scientific stakeholders.

## 5. Results

### 5.1. Public Hearings

Altogether, 420 people spoke up in the 50 meetings studied, with 20 being highest and four being lowest in a single meeting (Figure 1). Figure 2 depicts the spread of questions on different topics raised in these meetings. The maximum number of queries was raised on the topic of 'jobs related' followed by 'CER' requests. It implied that people were foremost interested in jobs and other benefits for the local community. Among the environmental impact issues, 'pollution' was the leading one. Within 'pollution', air pollution-related queries occupied the major share (Figure 3). 'Pollution control (without mentioning specifics)' came next, which indicated that, although people are concerned about possible pollution, they were not sure about its true nature and impact.

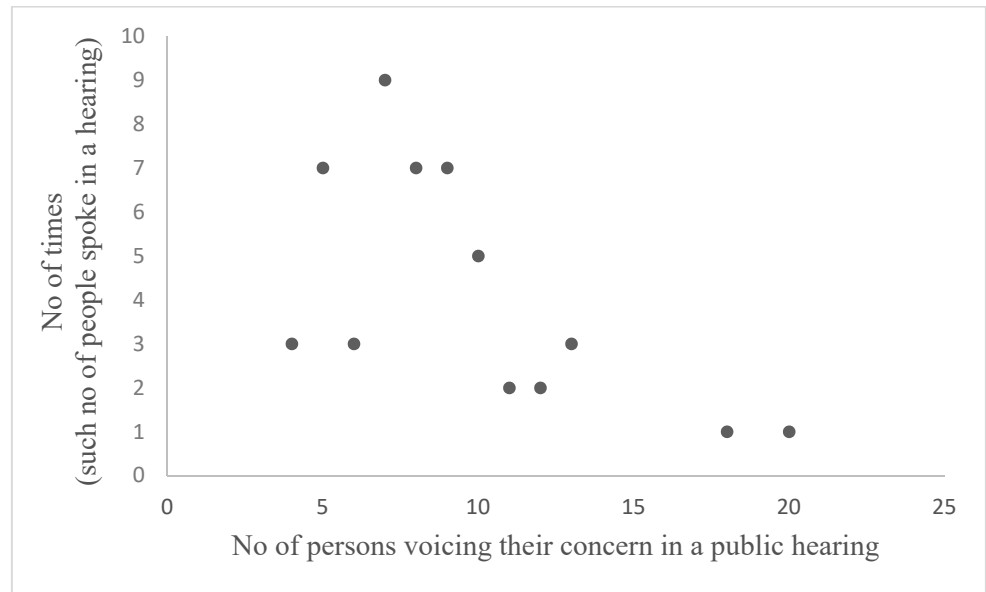

**Figure 1.** Number of persons speaking up in the 50 public hearings corresponding to 50 different proposed projects.

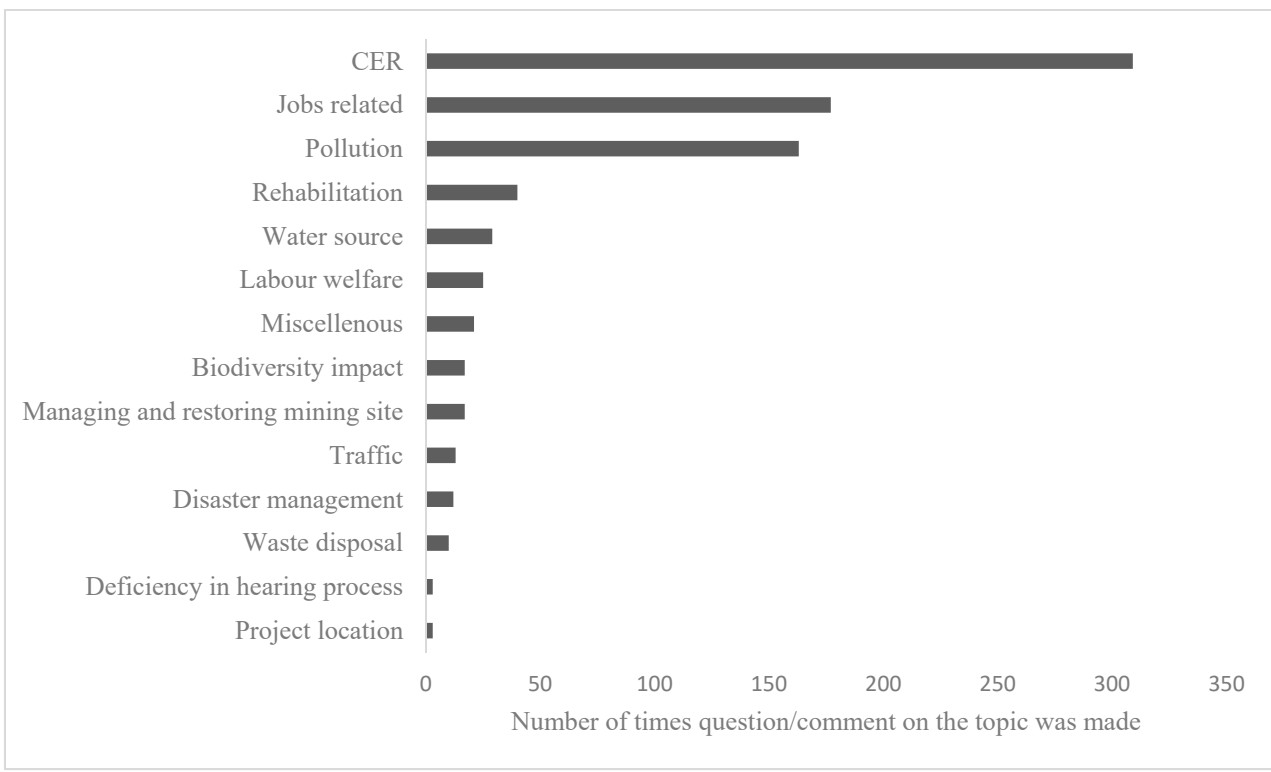

**Figure 2.** Spread of questions/comments from people on different topics in the 50 public hearings. Corporate Environment Responsibility (CER) is need-based activities for the local people which the project proponent is required to do around the project area.

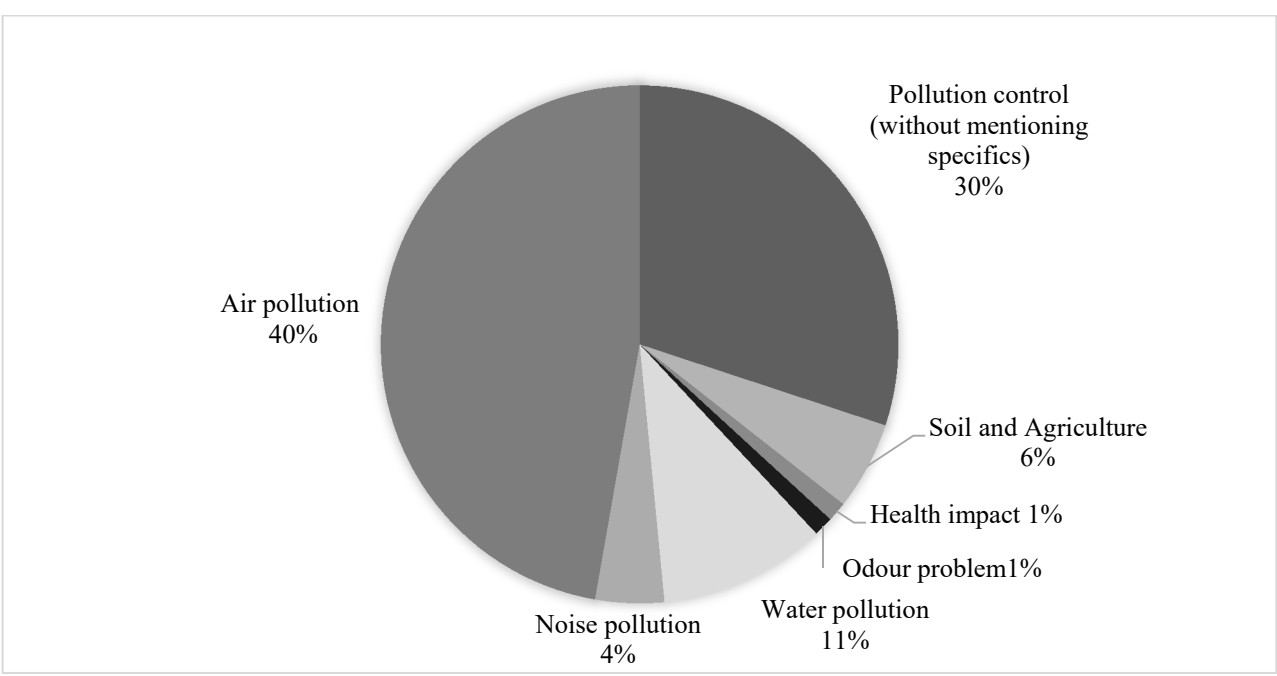

**Figure 3.** Pollution-related issues raised by people in 50 public hearings.

Among CER requests (Figure 4), plantation of trees found mention in almost half of the meetings and was foremost in terms of the volume of requests made by people. This indicated that people are still keen to improve green cover in their locality recognizing its importance to the environment. However, tree plantation requests did not contain

suggestions on species composition or the location of the plantation. There was no request on other aspects of biodiversity conservation/improvement.

A small number of comments was devoted to biodiversity impact issues. These issues were–forests should be affected by process activities; concern about deforestation; requirement of tree felling for the proposed project; details of compensatory afforestation; water pollution was causing the death of fishes in water stream; dissatisfaction about the tree plantation in an existing project (in which proponent at the time of hearing was proposing expansion); and, the loss of grazing land.

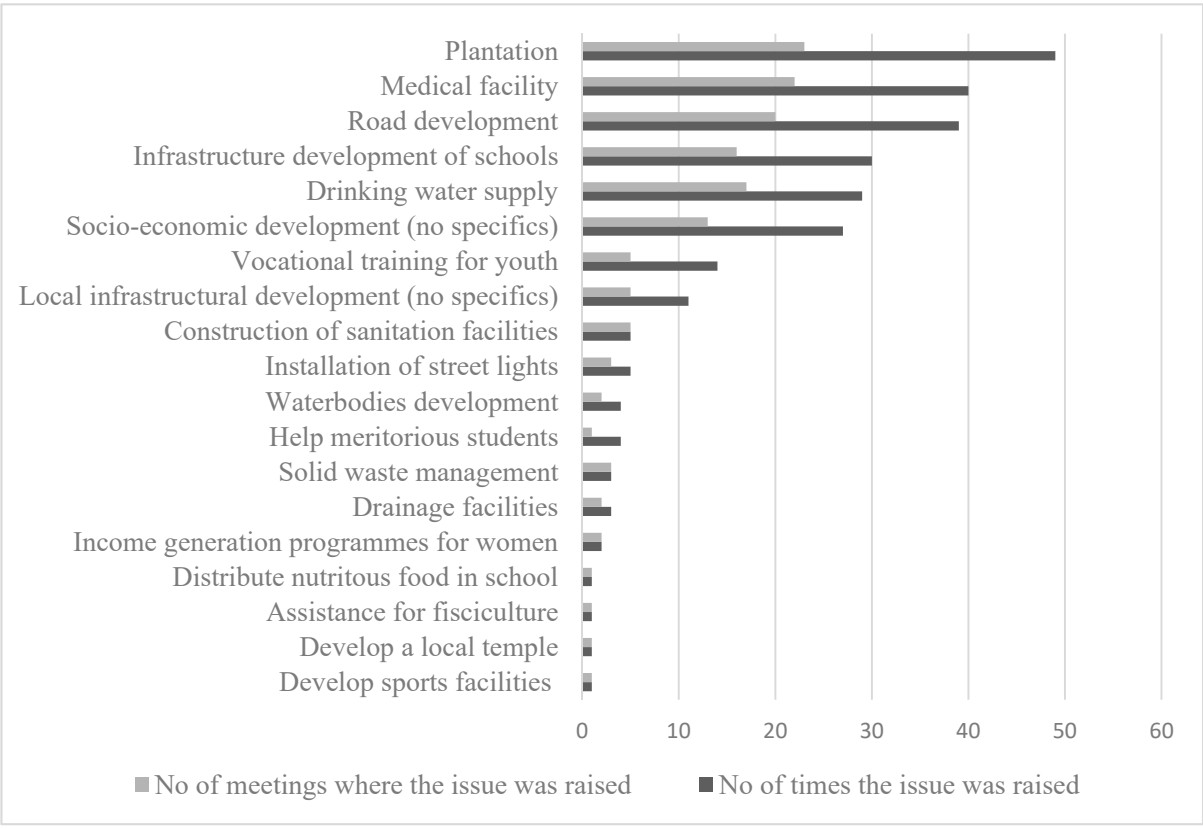

**Figure 4.** Topics on which requests were made to project proponents to spend under CER in the 50 public hearings studied. CER or corporate environment responsibility are need-based activities for the local people which the project proponent is required to do around the project area.

*5.2. EIA Reports*

All of the EIA reports (Table 1) provided baseline data containing an exhaustive list of flora and fauna within a 10 km radius around the project site. However, location-specific information like important habitats and the biodiversity-rich area was missing. Without such spatial baseline data, it would be difficult to predict impact or to make future conservation plans. Further, except for one report, none mentioned the dependence of people on bioresources and ecosystem services.

**Table 1.** Presence or absence of basic information on baseline data, impact assessment, and mitigation measures regarding biodiversity in the 20 EIA reports studied. Figure within bracket is the % of projects in which different items appear.

| Baseline Data | | | Impact Assessment | | | | Mitigation Measures | | | | | |
| --- | --- | --- | --- | --- | --- | --- | --- | --- | --- | --- | --- | --- |
| Name of species in the study area (10 km around the project location) | Location-specific information on biodiversity richness/ sensitive area | Dependence of people on biodiversity | Impact type | Specific area and amount of possible impact | Effect on bio-resources used by people | Mitigation measures other than tree plantation (in addition to mandatory pollution control measures) | Green belt development within the project site | | | Biodiversity enhancement activity proposed outside the project area. |
| | | | | | | | Plantation map | Name and number of tree species for plantation | Plantation schedule (with year-wise breakup of plantation and fund allocation for the purpose) | |
| Provided (100%). In two projects (10%) vegetation composition analysis (frequency, abundance, and density) was also done. | Absent. The only occurrence of protected forest within the study area was mentioned. | If the list of agricultural crops & medicinal plants is not counted, then only one project (5%) mentioned social forestry and the dependence of people on bioresources like wood and non-timber forest products. | All projects predicted insignificant impact. In 7 projects (35%), a general statement on the possibility of an impact on biodiversity from particulate air pollution due to industrial emission and transport was mentioned. | Absent | Absent | Absent | Absence of complete plantation map. In 6 reports (30%), green belt area was shown within project area but without scale and location of the species-specific plantation. | Total number of trees to be planted was mentioned in 18 projects (90%). A list of tree species was given in 19 projects (95%). The number for each species was given in 6 projects (30%). | Total fund allocation was provided in only12 projects (60%). Among them, the year-wise breakup was given in one project (5%). | In 11 projects (55%), tree plantation outside the project area was included under CER activities. But those proposals did not contain specifics of location, area, number, and species name for plantation. |

All of the reports inferred that there will be no significant impact primarily based on the non-occurrence of protected forest and endangered species within 10 km of the project site. Wherever mentioned, the impact statements were generic without the specifics of location and amount. None dealt with the loss of any bioresource or ecosystem service.

The only mitigation and offset measure that was proposed in all projects was green belt development within the project site. It may be observed that greenbelt development is a mandatory requirement as per terms of reference (TOR) for doing EIA and as per the West Bengal Trees (Protection and Conservation in Non-Forest Areas) Act, 2006. Thus, a proposal of greenbelt development was not an exclusive outcome of the EIA exercise. Further, plantation proposals were incomplete. Most did not have a plantation map showing a scaled dedicated area for greenbelt development within project land and plantation location of particular species based on canopy size.

In 11 reports, tree plantation outside the project area was proposed as a component of CER activities. However, no specifics regarding location, number, and species for plantation were provided.

While an exhaustive list of species (including the occurrence of rare species) demonstrated the scientific knowledge of the consultants preparing the reports; a lack of information on location-specific details and the use of bioresources by people indicated the scope of improvement in EIA reports through effective public consultation.

## 6. Discussion and Recommendations

The merits of a prevalent system in India are that public consultation is institutionalized through EIA Notification, 2006, and public hearings are conducted in the local dialect. However, there are two major deficiencies in legal provisions and prevalent practice. (i) In the public hearing, a presentation by the project proponent is followed by questions/comments from the public. Government officials chair the meeting and prepare the minutes, but do not take part in the main discussion. Thus, no subject expert can counteract or ask relevant questions to the project proponent for better public understanding about adverse impacts. (ii) Public consultation is only mandatory at the end of the EIA study.

Regarding the method on the assessment of the impact on biodiversity, there is, as such, no specific guideline in India. However, in recent years, the government has issued some sector-specific standard terms of reference (TOR), which seek information regarding biodiversity, like details of trees to be felled for the project, a description of flora and fauna existing in the study area with special references to rare, endemic, and endangered species, and comments of forest department if the project is to be located within 10 km of protected forest or migratory corridors of wild animals.

From this study, it was observed that, despite all existing legal provisions, public participation in EIA has not grown to its full potential. The study of minutes of public hearings and biodiversity-related chapters of EIA reports provided insights into the status of public participation after and during the preparation of EIA reports, respectively. Few people spoke in public hearings. The discussion was mostly on job opportunity and benefits; and, little on biodiversity impact issues. The possible reason for this attitude is that, being part of a developing nation with high population density, people are desperate for jobs.

In the public hearings studied, many raised the issue of pollution control, but without mentioning any specifics. These indicated that, although people were concerned about possible pollution, they were not sure about its true nature and impact. Therefore, educating the public and providing expert help before/during hearings are required for more meaningful participation. Inviting knowledgeable persons and NGOs may fill the knowledge gap to raise pertinent questions and enrich the discussion.

In India, as per the Biodiversity Rules, 2004, every local body has to constitute a Biodiversity Management Committee (BMC) whose main function is to prepare the People's Biodiversity Register (PBR) containing information on local biological resources and

their traditional uses. The documentation is done by collecting information from local people [36]. The near completion of all the PBRs in West Bengal (http://nbaindia.org accessed on 3 March 2021) indicates that there is a lot of knowledge and understanding on biodiversity issues within the local community. The lack of a simple method of BIA may be a reason that is restraining people from raising biodiversity impact issues in the hearings.

Apart from the mitigation measures of adverse impacts, the project proponent is required to undertake works around the project area for the betterment of the environment and people as part of CER. Suggestions that were made by people during hearings would be one of the bases for determining what these works would be. It was encouraging to find that tree plantation topped that list. However, there was no talk on other aspects of biodiversity conservation/improvement. Thus, there is a need to provide an easy guideline containing various (scientifically valid) options from which local people can choose based on local need and possibility.

In the EIA reports, a lack of information on location-specific details and the use of bioresources by people indicated inadequate public consultation during the EIA study. Several studies [3,7] recommended that stakeholder participation should be considered as early as possible and throughout the process. The EIA Notification, 2006, the legal instrument under which EIA is done, only provides for public consultation after the draft EIA has been submitted by the project proponent. This limits public participation to this late stage, rather than involving interested and affected parties earlier in the EIA. However, the aforesaid Notification provides for TOR (based on which the project proponent has to prepare the EIA report) whose content can be decided by the authority. Public consultation during EIA study can be ensured by making it mandatory to provide the details of persons and organization along with their opinion in the EIA report as well as information that requires public consultation (and field visit), like impact on bioresources that are used by local people. This could be achieved by the incorporation of additional requirements in the TOR for doing EIA.

The rapid loss of biodiversity in the tropics [37] makes every possible measure towards the protection of biodiversity on the ground very important and urgent. This work did not intend to undertake any systemic review of the literature. The purpose was to utilize the knowledge in existing reviews and relevant case studies to improve public participation in BIA. BIA is not a separate process, but a subset of EIA, in West Bengal because biodiversity is one of the many aspects considered under EIA. Therefore, an improvement of public participation in BIA can be achieved by increasing public participation in EIA in general together with people's ability to discuss biodiversity issues. A study of scientific literature on 'public participation in EIA' was done to find out the best practices of public participation in EIA and, subsequently, thinking of ways to implement them in West Bengal. The intention was to provide policy recommendations for increased public participation in EIA. On the other hand, the literature on BIA was studied to transfer knowledge to the public in a simplified format to increase their capacity to better assess the impact on biodiversity from a proposed project, thereby improving the quality of participation.

We identified four pillars of effective public participation in EIA (Figure 5). Specific recommendations under each of them for improving public participation in BIA in West Bengal are given below.

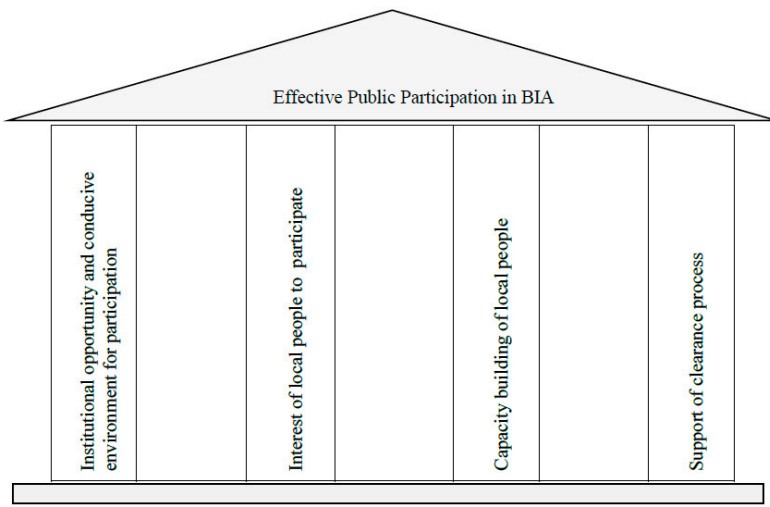

**Figure 5.** Four pillars of effective public participation in EIA.

*6.1. Institutional Opportunity and Conducive Environment for Participation*

Using scientific literature and issues that are raised in public hearings regarding process deficiencies, a list of relevant issues along with recommendations was prepared, as presented in Table 2. While most of those issues relate to the conduct of public hearing (like the location of hearings, language to be used, and publicity of hearing), but the list also included the issue of spreading public participation across the EIA process rather than a one-day public hearing. Although a single-day public hearing may not be enough, ordinary people cannot be expected to go to multiple meetings leaving their income-generating activities. Accordingly, for a specific project, the consultant of the project proponent should make field visits and talk to people. This can be ensured by incorporation in TOR to consult local organizations/persons during the EIA study. Additionally, some local organizations (like BMC or any educational institution) may take initiative in accumulating knowledge and opinion at the convenience of people. This knowledge may be used to list areas that are to be kept undisturbed and what needs to be done to enrich biodiversity in the particular region. Once the list and relevant map are prepared, these may be utilized by people during future hearings.

**Table 2.** Recommendations to provide opportunity and conducive environment for public participation in EIA.

| Sl No | Pertinent Issues | Source/Reference | Recommendation for Implementation |
| --- | --- | --- | --- |
| 1. | Public participation should be institutionalized | [7] | Based on the scale and pollution potential, a section of proposed activities statutorily require EIA and public consultation. Considering logistic and time constraint, it is difficult to it extends to the rest which also causes a cumulative adverse impact. Forming a regional plan containing 'no conversion green zones' through public consultation may be a solution. |
| 2. | Local dialect should be accepted in the meeting | [8] | Local dialect should continue to be used both by the project proponent/consultant and people. |
| 3. | A public hearing should be conducted near the project site as it would be easier for villagers to attend. | The issue was raised in the public hearings. | The public hearing is to be conducted in a convenient place near the project site easily accessible to the public. |

**Table 2.** *Cont.*

| Sl No | Pertinent Issues | Source/Reference | Recommendation for Implementation |
|---|---|---|---|
| 4. | Information should be given to participating stakeholders on time (a month before) to enable them to participate effectively. | [8] | Apart from the advertisement in two newspaper a month before the hearing (as presently done), publicity may be done through- i. putting posters in strategic public places and near project sites. |
| 5. | Wide publicity of hearing has to be undertaken. | The issue was raised in the public hearings. | ii. making loudspeaker announcements in the nearby areas 7 days before the hearing. |
| 6. | When decisions are highly technical, this may involve educating participants, developing the knowledge and confidence that is necessary for them to meaningfully engage in the process | [7] | Apart from distribution of simplified information among the public, other steps which may help are- i. some local organization like Biodiversity Management Committee may take initiative in understanding impact and become a knowledge exchange center ii. some knowledgeable persons, NGOs may be invited to the hearing |
| 7. | Stakeholder participation should be considered right from the outset, from concept development and planning, through implementation, to monitoring and evaluation of outcomes. | [7] | In addition to public consultation after preparation of draft EIA, the scope of public participation may be expanded by- i. making it mandatory to consult people who are likely to affect during the preparation of EIA ii. providing a mechanism for people to submit grievances regarding the implementation |
| 8. | Unless people are convinced that participation will involve some real influence over decision making, the public will be reluctant to participate | [6] | In the notice board of local government offices, the following may be exhibited- i. copy of signed minutes of the public hearing ii. submission of project proponent to the authority addressing the issues raised in public hearing and minutes of the relevant meeting of the authority. |
| 9. | The public can act as a manipulator or as a manipulation detector. Also, mobilized groups may monopolize public response. | [10] | Facilitating wide participation and making people understand that their own benefit is attached to the protection of the environment. |
| 10. | Stakeholder analysis needs to be done | [7] | Various stakeholders who may be affected should be identified and talked to during the EIA study. This is particularly relevant for the vulnerable section of the population. |

*6.2. Interest to Participate*

Some case studies [11,38] showed that a lack of interest among the public to participate can be a major hindrance. Suggestions for increasing public interest are:

- Raising awareness among people—efforts should be made to bring back the lost connection between people with nature. People should be reminded of existing biodiversity, benefits obtained from it, and their responsibility in protecting nature.
- Providing incentive—rural people are mostly poor and attending public hearings means losing a day's income. Hence, they may be compensated by non-monetary incentives, like environment-friendly products or nutritious food items.
- People should feel their opinion has value and can make difference [6]. The decision-making body should include in the meeting minutes that they are satisfied with the way the project proponent addressed the issues raised in public hearings. A place should be designated at a local level where people can register their grievances regarding the fulfilment of commitment by the project proponent.

*6.3. Capacity Building of Local People*

Different levels of public involvement are 'informing (one-way flow of information), consulting (two-way flow of information), or "real" participation (shared analysis and assessment)' [18]. The aim should be "real" participation for which we need to increase the capability of local people by providing simple tools. Here, we propose two such simple tools:

1   Matrix for determining biodiversity impact

Combining important features for biodiversity consideration in scientific papers, a matrix for determining adverse impacts was prepared (Table 3). The table contains project-related activities that may impact biodiversity (direct and indirect drivers) on the horizontal axis and biodiversity attributes on the vertical axis. An effort was made to keep it simple for ease of understanding and the use of local people. The matrix is expected to help local people (individually/collectively) to assess impact in a structured way and put forward their opinion efficiently. People can also use it as a checklist and seek the necessary information from the project proponent during public hearings. Important topics would be less likely to miss.

Intentionally, in the matrix, the only scope for assessing adverse impact has been kept so that undue stress on mitigation measures should not mask the adverse impacts. After assessing adverse impact, local people from their own field experience may judge whether the adverse impact could be avoided or reduced; or, whether the restoration and/or offsets proposals would work and mitigate the adverse impacts. For example, whether compensatory afforestation would take place at a similar scale (area and variety of life forms) or be as disjoint small fragments that can never sustain previous wildlife of an integrated forest.

2   Suggestive list of components for CER

Green belt development was the only proposed mitigation measure in EIA reports and tree plantation that topped the list of components that people demanded under CER. However, there exist various ways to conserve/enrich biodiversity. Scientific concepts from published papers for improving biodiversity are utilized to prepare a list (Table 4) from which local people can choose for placing demands under CER for the improvement of biodiversity in the locality. Besides, the project proponent can use the list for preparing offsets proposal to mitigate the residual impact.

**Table 3.** Table for use of local people for assessment of adverse biodiversity impact from a proposed project.

| Biodiversity Attributes | | Direct Drivers | | | | | | | | Indirect Drivers | |
|---|---|---|---|---|---|---|---|---|---|---|---|
| | | Land Character Change/ Land-Use Change | Pollution | | | Waste Dumping | Removal/ Extraction of Species/ Bioresources | Introduction of Species (Exotic Species/Invasive Alien Species/Genetically Modified Plant) | Other Project Activity which May Have Impact | Increased Access to Human Beings | Socio-Economic (e.g., Chance of Future Development and Land-Use Change in Future) | Cultural (e.g., Loss of Love and Reverence for Nature) |
| | | | Air | Water | Noise | | | | | | | |
| Concerning life-forms | Habitat loss | | | | | | | | | | | |
| | Disturbance to normal life activities, privacy | | | | | | | | | | | |
| | Number of species to decrease (species diversity) | | | | | | | | | | | |
| | Depletion of resources used by life forms | | | | | | | | | | | |
| | Lifecycle disturbance (migratory pathway, egg-laying spaces) | | | | | | | | | | | |
| | Disturbance to the food chain | | | | | | | | | | | |
| | Hazards to life-forms (plastic pollution, formation of deep wells in which animals may fall) | | | | | | | | | | | |
| Resilience | Loss of functional diversity and loss of indigenous species | | | | | | | | | | | |
| | Loss of mobile link species (birds, small mammals) | | | | | | | | | | | |
| | Fragmentation of habitat | | | | | | | | | | | |

**Table 3.** *Cont.*

| Biodiversity Attributes | | Direct Drivers | | | | | | | | Indirect Drivers | |
|---|---|---|---|---|---|---|---|---|---|---|---|
| | | Land Character Change/ Land-Use Change | Pollution | | | Waste Dumping | Removal/ Extraction of Species/ Bioresources | Introduction of Species (Exotic Species/Invasive Alien Species/Genetically Modified Plant) | Other Project Activity which May Have Impact | Increased Access to Human Beings | Socio-Economic (e.g., Chance of Future Development and Land-Use Change in Future) | Cultural (e.g., Loss of Love and Reverence for Nature) |
| | | | Air | Water | Noise | | | | | | | |
| **Ecosystem services** | Source of water/food | | | | | | | | | | | |
| | Source of articles of use | | | | | | | | | | | |
| | Source of livelihood (especially of vulnerable section) | | | | | | | | | | | |
| | Source of other ecosystem services (pest control, pollination, prevention of soil erosion, flood control) | | | | | | | | | | | |
| | Landscape/sacred grove/heritage site | | | | | | | | | | | |
| **People's attitude** | Interest of people to save the biodiversity (including economic interest) | | | | | | | | | | | |

Keeping in the mind the factors of magnitude, reversibility, scarcity, and possibility of impact, the adverse impact may be marked as high (+++), moderate (++), low (+), and in case very low or no adverse impact the space to be left vacant. If anyone feels something important will be lost, details may be given under the following topics-Habitat; Species; Ecosystem services.

**Table 4.** A suggestive list of components from which people may choose to request project proponents to spend under CER for biodiversity conservation/improvement.

| Sl. No. | Concept | Source/Reference | Ways of Implementation |
|---------|---------|------------------|------------------------|
| 1. | Creating wild patches (which will work as stepping stones) | [23,26] | i. Conversion of some agricultural land/wastelands into natural ecosystems such as wild patches.<br>ii. Small area of government land/public land where local indigenous plants may be planted.<br>In these patches, a variety of indigenous trees/shrubs would be planted and kept undisturbed. These will not only serve as local repositories of biodiversity but also a habitat for organisms (e.g., beehives), and subsequently become a source of ecosystem services. |
| 2. | Maintaining wild patch and protection of natural habitat | [27,28] | i. Protection of existing wild patches which are acting as important habitat (including waterbodies visited by migratory birds).<br>ii. Protection of sacred grooves. |
| 3. | Connecting links | [23,29] | Plantation along roads and irrigation canals. Apart from avenue trees, fruit-bearing trees may also be planted which may serve as a food source (option value) at times of crisis. |
| 4. | Increase heterogeneity of the landscape | [23,26] | i. Creation of waterbodies.<br>ii. Creation of patches of trees among agricultural fields. |
| 5. | Keystone species | [17] | Plantation of trees like *Ficus bengalensis*, *Ficus religiosa.* |
| 6. | Plantation of indigenous trees and conservation of locally cultivated varieties | [30,39] | i. Plantation of indigenous trees/shrubs in common land and along roads.<br>ii. During EIA study, a list of local flora is prepared. Tree species may be selected from this list for plantation within the premises (green belt).<br>iii. Providing an incentive for the cultivation of traditional varieties of crops. |
| 7. | Support for mobile link species | [16] | Birds are important seed dispersers. In order to support them, fruit-bearing trees like *Mimusops elengi*, *Ficus religiosa*, *Syzygium cumini*, *Azadirachta indica* may to be planted. |
| 8. | The control of invasive species | [33] | Executing eradication programs for invasive species like *Parthenium hysterophorus.* |
| 9. | Waste management and recycling of nutrients | [24,40] | i. Plastic waste management (so those plastic items are not consumed by animals)<br>ii. Composting facilities to be developed and compost manure to be put in the field for nutrient recycling |
| 10. | Increase the interest of local people to protect biodiversity | [12] | People will most likely protect biodiversity if they feel that they are being benefitted from it. Through awareness campaigns, locally existing biodiversity and ecosystem services provided by it should be explained to people with examples. |

*6.4. Support of Clearance Process*

There can be elements ingrained in the clearance process, so that public consultation becomes essential for project proponents during EIA study and public opinion expressed in public hearings gets due importance during the appraisal by sanctioning authority. Some suggestions are:

(a)    Mandatory requirements in EIA

The following may be mentioned in TOR for inclusion in the EIA report:

• Details of local organizations and persons consulted during the EIA study and their opinion should be mentioned.

- Location-specific information on biodiversity-rich areas and habitats should be provided.
- The impact statement should not only talk about endangered species, but include common species and their habitats.
- Impact on bioresources that are used by people and other ecosystem services to be mentioned.

(b) Authority should peruse the issues raised in public hearings—project proponent should submit a synopsis of how the issues raised in public hearings have been addressed. The decision-making body should include in the meeting minutes that they are satisfied with the way the project proponent had addressed the issues.

(c) Mechanism for monitoring of implementation—there may be a committee at a local level consisting of members from the local government, the Biodiversity Management Committee (which contains representatives from the local community), and the project proponent who should verify the implementation of commitments and report to the authority.

## 7. Conclusions

EIA documents are prepared by consultants that were employed by the project proponent. In most cases, the sanctioning authority is not even visiting the project site and taking decisions based on the documents that were submitted by the project proponent. Environmental consultants are expected to have rich scientific knowledge that can be supplemented by local knowledge from the public [5,41]. Further, public scrutiny can act as a possible check on the authenticity of the documents [5]. Proceedings of public hearings can become a valuable second channel of information for decision-making authority. However, public hearings are not free from loopholes. Chances of various types of manipulation exist, especially in the absence of wide participation [10]. For all of these reasons, public consultation during and after the preparation of EIA reports is desirable, not only in terms of quantity, but also quality.

We provided recommendations for improving public participation by taking a holistic approach. Suggestions have been provided covering the entire process for authorities to consider. Because impact issues of the proposed project and components under CER are the two topics on which people need to speak in public hearings, we tried to enhance their capacity by proving simple tools for both. The simple matrix would help local people to seek relevant information and assess impact. The suggestive list of components for improving biodiversity would provide a range of options to choose from while putting forward demands under CER. The implementation of the recommendations stated in this paper could help in improving public participation concerning BIA. Although being developed with a focus on West Bengal, the recommendations are not limited in applicability to India, but may be suitably used in other parts of the world.

**Author Contributions:** Conceptualization, R.C.; methodology, R.C.; formal analysis, R.C.; investigation, R.C.; resources, R.C.; data curation, R.C. and W.W.; writing—original draft preparation, R.C.; writing—review and editing, A.L. and W.W.; visualization, R.C. and A.L.; supervision, W.W.; project administration, A.L. and W.W.; funding acquisition, R.C. and A.L. All authors have read and agreed to the published version of the manuscript.

**Funding:** This work was done as a part of the fulfillment of the 43rd UNEP/UNESCO/BMU International Postgraduate Course on Environmental Management for Developing Countries in which the first author participated at CIPSEM of the Technische Universität Dresden receiving a full scholarship from the German Ministry for the Environment, Nature Conservation and Nuclear Safety (BMU). Open Access Funding by the Publication Fund of the TU Dresden.

**Acknowledgments:** We would like to thank Anna Görner and her colleagues at Technische Universität Dresden, Faculty of Environmental Sciences, Centre for International Postgraduate Studies of Environmental Management (CIPSEM) for their support and encouragement in the completion of the work.

**Conflicts of Interest:** The authors declare no conflict of interest.

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
