# Peer review of "Public Participation in Biodiversity Impact Assessment in the State of West Bengal, India: Present Status and Finding Ways for Improvement"

_environments, doi:10.3390/environments8050039_

Round 1
Reviewer 1 Report
The authors presents an interesting research, based on the public participation in EIA studies, focusing in the biodiversity approach. After checking 50 public hearings and 20 EIA reports, they conclude that the most showed interest was in potential jobs and direct benefits. Few items on biodiversity impacts were recorded. Moreover, the authors discuss about of effective public participation identifying basic pillars.
The manuscript fits perfectly with the scope of the journal and I’m sure that will arise the interest of a large audience but right now it should be restructured notably.
- The main thematic of the manuscript is not clear. Which are the objectives? Which are the questions that your research wants to answer?
- The manuscript starts on the study of scientific literature regarding public participation and BIA concerns on EIA but this topic finish there.
- After, in methodology block, the authors refer to public hearings and the study of EIA reports in West Bengal, but also on public participation on 10 (randomly) developing countries. From this analysis, a set of recommendations is arisen. I think that this methodology is not appropriate, specially the random selection of developing countries to compare.
- Results refers to public hearings and EIA reports focusing on the biodiversity impact issues. After this, appears the analysis of public participation in developing countries but is not well related with the previous West Bengal analysis.
- At the end, the discussion and recommendations block are devoted to four pillars of public participation but disconnected of the goal of the research.
- Finally, the conclusions are not clear and do not respond the objectives of the research.
Without doubt, the authors have been working hard analysing a lot of information and there are a lot of material to be published in Environments, but a deep restructuring is needed in order to become a paper.
My recommendation is to focus in one aspect (i.e. public auditions and biodiversity or public participations on EIA), rewrite the manuscript and send another time.
Author Response
Dear Reviewer 1,
thank you very much for your constructive feedback; please find our detailed response attached.

Reviewer 2 Report
Interesting and relevant paper, but so confuse in its actual redaction dan lack of several explanations.
Remarks
Line 64. These are -?
Line 97. “the relevance of the paper’s title” What means? Which criteria was used to determine the title relevance?
Section 3.2. BIA and biodiversity conservations.
The Authors talk about “The literature on BIA”. This section is review of BIA literature or not? If it is, which criteria was used to select the cited papers?
Section 4.1. Study of minutes…
Why the selection of 50 hearings? This number is representative for West Bengal?
Line 148. “Comments made in the hearings…”. Who collected this type of information, and how? This work was done every time for the same people? Were the hearings recorded and then analyzed, or the topics from hearings was “hand-recorded”? She, He, They used a previous established topic table as a guidance? Which and how items/topics were previously selected /defined?
Line 161. Which indicators were used to developing countries selection? And why 10?
Line 173. About 8 people by hearing (420/50) is representative /relevant in relationship with the community size?
Figure 1 and 2. Are there any relationship between Figure 1 and 2. Are the same items?
Figure 2: Legend for the axis is disappear.
Figure 3. How a figure only for Pollution related issues?
Line 194. “….existing plant…” means?
Table 1. Include % of projects in which different items appears may be useful to show the level of representation.
Line 234. “Important provisions…” Important legal provisions…?
Line 269 . ToRs means? Is the same TOR (Line 317)
Paragraph Line 318-331. These ideas/objectives were explained before, it don´t add new/relevant information. information was
Figure 5. These information/conclusions were extracted from the analysis of hearings, from the bibliography review…?
Section 6.1 I don´t understand the relationship between Table 3 recommendations (form a bibliography review) and the “Ten developing countries” information analysis about EIA procedures.
Table 4. Please, explain how the local people could use Table 4.
Author Response
Dear Reviewer 2,
thank you very much for your constructive feedback; please find our detailed response attached.

Round 2
Reviewer 1 Report
After reading the authors responses The mauscript is ready for become a paper in Environments journal. The authos have improved notably the manuscript. Congratulations.
Reviewer 2 Report
Changes have done in a properly way.